

# Quantifying changes in seasonal temperature variations using a functional data analysis approach

Eva Holtanová[1], Jan Koláček[2], Lukas Brunner[3]

[1]Department of Atmospheric Physics, Faculty of Mathematics and Physics, Charles University, V Holešovičkách 2, Prague, 180 00, Czech Republic
[2]Department of Mathematics and Statistics, Faculty of Science, Masaryk University, Kotlářská 267/2, 611 37, Brno, Czech Republic
[3]Research Unit Sustainability and Climate Risk, Center for Earth System Research and Sustainability (CEN), University of Hamburg, Hamburg, Germany

*Correspondence to*: Eva Holtanová (eva.holtanova@matfyz.cuni.cz), Lukas Brunner (lukas.brunner@uni-hamburg.de)

**Abstract.** Ever-worsening climate change increases near-surface air temperatures for almost the entire Earth and threatens living organisms and human society. While annual mean changes are frequently used to quantify past and expected future changes, the increase is rarely uniform throughout the year. In addition, the shape of the annual cycle and its changes can differ considerably between regions around the globe. Therefore, we perform a global analysis resolving the annual cycle and its changes in different regions, focusing on diagnostics that can be evaluated for various existing annual cycle shapes (e.g., single and double waves, different timing of seasons, etc.). Many previous studies relied on parameter-based methods, assuming a sinusoidal shape of the mean annual cycle. Here, we introduce an innovative approach based on Functional Data Analysis (FDA), a relatively new statistical approach. The evolution of the mean annual cycle is estimated from daily long-term mean temperature values, which are converted to functional form. We concentrate on diagnostics that evaluate the change in absolute temperature, its seasonal slope, the position of the maximum, and the amplitude of the annual cycle. We analyze two reanalysis datasets (coupled CERA20C and atmospheric ERA5) and a subset of five CMIP6 Earth system models (ESMs). Observed changes in the second half of the 20th century are assessed, and the ability of ESMs to represent them is evaluated. Further, the changes projected for the end of the 21st century under the SSP3-7.0 pathway are analyzed. Among other results, we highlight distinct differences between the two reanalyses, especially over equatorial and polar regions across diagnostics. Our approach also reveals that differences in the historical period between 1951-1980 and 1981-2010 can be negative during (short) parts of the year in many regions. Further, the ESMs future projections show different rates of warming between seasons, resulting in changes in the amplitude. The largest amplitude increase is projected over the Mediterranean region, and the largest decrease over the Arctic Ocean, the latter being due to the considerably stronger warming in the northern hemisphere winter. The ESMs also project a delayed maximum near the poles and an earlier maximum in many tropical continental regions. In



Europe, the southern and eastern regions experienced a delay of the maximum of up to 10 days, whereas a slightly earlier
maximum is found for northern Europe. A similar dipole pattern can be seen between eastern and western regions in North
America. Regarding the slope of the annual cycle, higher latitudes detect a higher magnitude of change in the historical period
than lower latitudes. The geographical pattern remains the same for future slope changes, with the magnitude twice as high in
most regions. The FDA diagnostics introduced here can be tailored for different purposes and applied to other climatic
variables, without making any prior assumptions about the annual cycle shape. Potential applications include, e.g., explicitly
evaluating the climate model performance or ensemble mean and spread assessment beyond annual or seasonal means.
**1 Introduction**
Increasing near-surface air temperature is observed and projected for almost the entire Earth (IPCC, 2021), threatening the
environment and human society alike. However, this temperature increase is rarely uniform throughout the year, and even if
the annual mean changes only slightly, the annual cycle might change quite dramatically (Marvel et al., 2021; Wang et al.,
2021). The changes in seasonal temperature cycle can have potentially large impacts on, e.g., phenological phases of living
organisms, agriculture, health, tourism, and other sectors. Widespread expected changes in the annual cycle have even
motivated suggestions of new definitions of seasons (Wang et al., 2021; López-Franca et al., 2022). Moreover, as noted by
McKinon and Huybers (2024), the shape of the temperature annual cycle can be taken as an analogy for temperature changes
in general, as it is easily distinguishable from internal variability and can be reliably observed. They emphasize that the
seasonality of temperature in the current climate and its changes are strongly related to projected temperature changes, and
recent changes in the mean annual cycle can be considered a proxy for overall future warming. The skill of climate models in
depicting correctly the observed shape of the annual cycle and its changes is therefore very informative in terms of confidence
in simulated future changes (Lynch et al., 2016).
A large number of previous studies have shown that the shape of the temperature annual cycle has already changed in recent
decades, including, e.g., a phase shift towards an earlier onset of the seasons over the middle and higher latitudes (evaluated
using the sinusoidal approximation of the mean annual cycle shape, the results do not relate to a specific season, Stine et al.,
2009), lengthening of summer (Park et al., 2018) and shortening of all other seasons over Northern Hemisphere midlatitudes
(Wang et al., 2021). Wang and Dillon (2014) revealed regionally different changes of annual cycle amplitude over northern
hemisphere midlatitudes and polar regions, with a prevailing decrease in 1975-2010 in comparison to 1961-1990. In addition
to adaptation to recently observed shifts, it is also crucial to investigate the expected future evolution, as the shape of the annual
cycle is expected to undergo even more dramatic changes during the upcoming decades. For example, Santer et al. (2018,
2022) found an increase in the temperature amplitude globally and throughout the troposphere in recent observations and future
projections and attributed it to anthropogenic forcing. Further, Chen et al. (2019) concluded that the CMIP5 global climate
models project increased seasonal amplitudes in low-latitude regions and most global ocean areas. In contrast, the seasonal
amplitudes are expected to decrease over the Southern Ocean and high-latitude regions.



Earth system models (ESMs) are state-of-the-art instruments for assessing possible future climate evolutions and attributing
observed and projected climate changes to their potential causes. The multi-model ensemble produced under the Coupled
Model Intercomparison Project Phase 6 (CMIP6) initiative, coordinated by the World Climate Research Programme's (WCRP)
Working Group on Coupled Modelling (Eyring et al., 2016), represents the newest set of ESM simulations. This ensemble
includes simulations of a range of different models under several shared socioeconomic pathways (SSPs, Tebaldi et al., 2021),
enabling the analysis of uncertainties arising from structural model differences (Abramowitz et al., 2019). Indeed, despite
indisputable progress in the complexity of the newest generation ESMs, many uncertainties and issues still need to be solved
(Shaw and Stevens, 2025; Randall et al., 2019; Bordoni et al., 2024). The problem of the choice of ESMs appropriate for
climate change scenarios is a very complex task, and different approaches are still under investigation (e.g., McDonnel et al.,
2024; Snyder et al., 2024; Merrifield et al., 2023;  Rahimpour Asenjan et al., 2023).
The shape of the annual cycle of air temperature differs significantly among different regions around the globe. Therefore, a
global analysis requires focusing on quantities that can be evaluated for all these different shapes (e.g., single and double
waves, different timing of seasons, etc.). A lot of previous studies relied on Fourier-transform-based methods, assuming a
sinusoidal shape of the mean annual cycle and focused on its amplitude and phase (e.g., in Paluš et al., 2005, Stine et al., 2009,
Zhao et al., 2021, Marvel et al., 2021, Deng and Fu, 2023, Zhang et al., 2025), which in some cases resulted in omitting certain
regions (e.g., Dwyer et al., 2012, Yettella and England, 2018) and a large portion of studies focused on the northern hemisphere
only. Here, we introduce an innovative approach based on Functional Data Analysis (FDA). The evolution of temperature
throughout the year is approximated by daily long-term mean temperature values, which are converted to functional form (see
Section 3 for details). This approach allows us to assess any existing shape of the temperature seasonal cycle. López-Franca
et al. (2022) also employed smoothing of daily temperature values with splines and evaluated changes in dates of minimum
and maximum of the smoothed annual cycle and the dates of minimum and maximum slope changes. Unlike the methodology
presented here, they only concentrated on specific parts of the year and the midlatitude regions. We previously successfully
applied a Functional data analysis approach to investigate the influence of driving the global climate model on nested regional
climate simulation within a multi-model ensemble (Holtanová et al., 2019).
**2 Data**
The present study deals with the mean annual cycle of near-surface air temperature. To analyze its recent changes over both
land and ocean regions, we use two reanalysis datasets from the European Centre for Medium-Range Weather Forecasts
(ECMWF), namely the ERA5 (Hersbach et al., 2020) and CERA20C (Laloyaux et al., 2018). The choice was motivated by
the long temporal coverage of these datasets back to the 1950s. Some basic information about these datasets is described in
Table 1. One of the main differences between them is that ERA5 is an atmospheric reanalysis; in contrast, CERA20C was
created using a coupled modeling system with the representation of not only the atmosphere but also the ocean, land, oceanic
waves, and sea ice. The atmospheric modeling system (ECMWF's Integrated Forecast System (IFS) version CY41R2) is the



same for both ERA5 and CERA20C (Laloyaux et al., 2018; Hersbach et al., 2020). As the coupling demands large
computational costs, CERA20C has a coarser horizontal resolution (Tab. 1). The CERA20C dataset includes 10 members
representing the spread related to the errors in the assimilated observations and the modeling system (Laloyaux et al., 2018).
We use the "number0" ensemble member and do not analyze the uncertainty spread here.
Further, we select historical and scenario simulations of five CMIP6 ESMs (Table 2). The model choice is motivated by the
different values of the equilibrium climate sensitivity (Meehl et al., 2020) and overall good performance compared to the whole
CMIP6 ensemble (Bock et al., 2020). We employ only five models to be able to analyze the individual simulated curves of the
mean annual cycle and illustrate the innovative methodology properly. For the scenario period, we analyze outputs for the
SSP3-7.0 socio-economic pathway, representing the medium to high end of the whole range of the SSPs currently considered
plausible (Tebaldi et al., 2021).
The analysis focuses on the periods described in Table 3. The two historical periods are used to assess recent observed changes
(Section 4.1). The difference between the future and reference periods is the projected or expected future change (described in
Section 4.2). For both the reanalyses and ESMs, the long-term mean values of near-surface air temperature for each day of the
year are averaged over the reference regions from Working Group 1 of the IPCC AR6 (Iturbide et al. 2020) directly from the
native grids. These daily long-term mean values are then subject to functional data analysis as described in the following
section.

## 3 Functional data analysis approach

### 3.1 Construction of the functional data

The modeling of the mean seasonal cycle of temperature uses the techniques of Functional Data Analysis (FDA), a relatively
novel statistical approach (Ramsay and Silverman, 2005; Horváth and Kokoszka, 2012; Kokoszka and Reimherr, 2017). Unlike
traditional statistics, a single observation of a variable is not a data point but rather a function. This approach is especially
suitable for a series of observations with an underlying correlation structure.
Generally, the relation between a covariate $x$ and a response $Y$ can be modeled as a function $y = f(x)$ using the data pairs $(x_i,$
$Y_i)$ with $i = 1,...,n$. In our case, the covariate $x$ is represented by the days of the year ($x_i$ varies from 1 to 365, for leap years,
values for February 29 were deleted). The mean seasonal cycle of temperature plays the role of response $Y$. To account for the
periodic nature of the data, the function $f(x)$ is defined as a linear combination of Fourier basis functions:
$$f(x) = a_0 + \sum_{n=1}^{m} \left( a_n \cos \cos \frac{n\pi x}{365} + b_n \sin \sin \frac{n\pi x}{365} \right) \tag{1}$$

i.e., $f(x)$ depends on $K=2m+1$ coefficients $\{a_0, a_1, b_1, ..., a_m, b_m\}$ and basis functions (see Fig. 1a for $K=5$).
The coefficients $\{a_0, a_1, b_1, ..., a_m, b_m\}$ are chosen to minimize the following functional:
$$\sum_{i=1}^{n}[Y_i - f(x_i)]^2 \tag{2}$$





Our approach fits a function f with varying degrees of freedom to the data. In contrast to a simple interpolation, this does not
necessarily mean that the function just connects all adjacent data points (i.e., in all cases where the degrees of freedom are less
than the number of data points, see Fig. 1). In general, the particular values, $x_i$, of the covariate and the corresponding observed
responses, $Y_i$, are linked by $Y_i = f(x_i) + \varepsilon_i, \ i = 1,...,m,$ where εi are realizations of the random errors. This corresponds to the
situation where the covariate, $x_i$, is given, and the observed response, $Y_i$, is the realization of some random variable linked with
the value of $x_i$. The resulting function f balances the size of the errors, $\varepsilon_i$, and the smoothness of the function linking the
covariate and the response. The smaller the number of basis functions $K$ is, the less sensitive it is to fluctuations in the data –
compare panels (b) and (d) in Fig. 1 for cases $K=5$ and $K=55$. Here we choose $K=15$. The choice is supported by the fact that
for this value, the character of the FDA curve best resembles the 30-day running average. The 30-day average is analogous to
the monthly mean, and the length of the month is an intuitive choice in climatology, as generally a lot of climatological analysis
is based on monthly mean values. Moreover, even for $K=5$, the FDA function explains more than 99% of the variance of the
30-year mean temperature values, even though the curve does not entirely align with the underlying data (Fig. 1(b)). On the
other hand, for higher K, the curve becomes too fluctuating, resembling high inter-daily variability in the data. Therefore, we
consider smoothing based on K=15 appropriate for the current study. However, the analysis results are not sensitive to the
choice of K (not shown). The FDA-smoothed curves of the mean annual cycle for all the datasets and geographical regions are
shown in the supplemental Fig. S03 and S07 for the historical periods, and in Fig. S04 and S08 for the projections.
**3.2 FDA diagnostics**
Drawing on the FDA representation of the annual cycle in each period specified in Table 3, we now define diagnostics that
evaluate changes in the shape of the annual cycle (sections 3.2.1 - 3.2.5). Fig. 2 illustrates the interpretation of the diagnostics
on example data. Table 4 provides an overview of the diagnostics and references to the figures showing the results based on
them. All the diagnostics are further used to quantify differences between the annual cycle curves in the two historical periods
and between the future and reference periods. For the historical periods, we compare the GCMs with ERA5 and CERA20C.
For future time periods, we compare individual GCMs with their multi-model mean. We want to emphasize that in the
projections, the multimodel mean values are based on the multimodel mean annual cycle, not the multimodel mean of the FDA
diagnostics. Therefore, the multimodel mean values of FDA diagnostics can fall outside the range of individual ESMs.
**3.2.1 Annual cycle shape**
For each day of the year, we calculate the distances between the smoothed annual cycle curves (see Fig. 2 (a)). We then
aggregate these distances in three ways: by calculating the 10th and 90th percentiles and the root mean square of them. The
former two diagnostics, hence, represent high and low annual extremes of the temperature changes (allowing both positive and
negative values), while the latter diagnostic evaluates the Euclidean distance of the whole annual cycles (positive by definition).
Supplemental Fig. S01 and S02 show the occurrence of values below/above 10th/90th percentiles; the red dashed line



represents the 10th percentile, and the green dashed line represents the 90th percentile. For all values below/above the 10th/90th
percentile threshold, the time periods of the year when these values occur are shown in red/green.
**3.2.2 Annual cycle maximum**
We define the shift in the annual cycle maximum as the number of days between the maximum of the annual cycle in two
periods (black arrow in Fig. 2(b)). Positive values indicate a delay in the maximum occurrence relative to the reference period,
and negative values vice versa. In regions with two (local) maxima in the annual cycle, the "first" and "second" maxima are
considered chronologically from January 1st, with no regard to the actual maximum magnitude (the second maximum can
potentially have a higher temperature than the first). There are nine regions, where we identify two distinct maxima, see Fig.
5, 6, S03, S04.
**3.2.3 Annual cycle velocity**
We calculate 1st derivative of the smoothed curve of the temperature annual cycle. We define temperature velocity as the
absolute value of this 1st derivative curve. It gives an indication of the steepness of the annual cycle on individual days (see
Fig. 2 (c)). Then we calculate changes in temperature velocities between corresponding days of the year between the two time
periods. Positive differences in temperature velocity, hence, indicate days where the annual cycle is getting steeper compared
to the reference period, and vice versa for negative values. Note that steepness means faster warming as well as faster cooling
because we consider absolute values of the 1st derivative. Similarly to the changes in temperature itself (3.2.1), we calculate
the 10th and 90th percentiles of the differences and their root mean square.
**3.2.4 Annual cycle amplitude**
The amplitude of the annual cycle is defined as the difference between the maximum and the minimum value in °C (see Fig.
2 (d)). Here we evaluate the change in the amplitude between two time periods. Consequently, a positive change in the
amplitude indicates an increasing temperature range over the year compared to the reference.
**4. Results**
Here, we discuss changes in the four diagnostics from a high-level perspective; detailed figures for each of the regions can be
found in the Supplement (see Table 4 for an overview).
**4.1 Changes in the shape of the annual cycle**
In the often employed annual-mean view, warming is evident almost everywhere on the globe, with land areas and higher
latitudes generally warming faster (Gulev et al., 2021). Our approach resolves seasonal differences in the long-term warming
signal and reveals that differences in the historical period between 1951-1980 and 1981-2010 can be negative during (short)



parts of the year in many regions (Fig. 3b). This is compensated by a strong warming in other parts of the year, which can
exceed 2 °C in many northern hemisphere land regions (Fig. 3c).
Fig. 3a shows the resulting aggregated differences in the shape of the seasonal cycle as the root-mean-square of the daily
differences (RMSD, also termed Euclidean distance), which also exceed 1.5 °C in most datasets at northern mid-latitude land
regions. In most other parts of the world, except Antarctica, the RMSD remains lower than 1.5 °C for the historical periods.
We stress that this diagnostic embraces both negative and positive temperature changes, evaluating the overall change in the
shape of the annual cycle, unlike simply averaging the changes over the year.
With regard to warming, in general, a stronger signal is seen in the northern hemisphere than in the southern hemisphere. An
exception is the strong warming signal in CERA20C in Antarctica (Fig. 3). Larger disagreement between the reanalyses also
occurs over the southern ocean and in some regions near the equator (e.g., SAH and ARP).
With regard to the timing, both reanalyses show the highest temperature increase during northern-hemisphere winter, or the
changes do not have any distinct maximum/minimum (Fig. S01). Only in New Zealand (NZ), the Southern Ocean (SOO), and
Antarctica (WAN and EAN) are the changes larger in the southern-hemisphere winter. In the ESMs, the timing of the largest
increase/decrease often does not match the reanalyses  (e.g., over Greenland, the reanalyses show a decrease of temperature in
the first three months of the year, whereas the ESMs show the decrease (if any) later in the year, Fig. S01). In the Arctic region
(ARO), the ESMs and reanalyses generally agree that the lowest increase in temperature occurs in summer.
For the warming at the end of the 21st century, the Arctic stands out with temperature increase exceeding 10 °C in all models
during the 10% of strongest warming days (Fig. 4c). Such stronger warming in the polar regions compared to lower latitudes
(often referred to as polar amplification) is consistent with theoretical considerations and historical observations (e.g., Stuecker
et al., 2018; Previdi et al., 2021). Here, we show that the stronger warming at high latitudes predominantly comes from the
upper end of the annual temperature distribution, with the 10th percentile of changes being mostly uniform across latitudes
(Fig. 3b and 4b).
Polar amplification has also been reported to be underestimated in CMIP6 models (Casado et al., 2023) and to be weaker in
Antarctica than in the Arctic region in both observations and CMIP6 models (e.g., Zhang et al., 2023; Xie et al., 2022). Our
results contradict these results to some extent. Mainly, the five ESMs simulate the magnitude of historical warming in the
Arctic, higher or comparable to the reanalyses (Fig. 3). Finally, we note that in the historical period, one of the two observation-
based reanalyses, CERA20C, shows stronger warming in Antarctica than in the Arctic, contradicting. This discrepancy might
be attributable to high decadal variability in Antarctica (Casado et al., 2023) and large uncertainties of the reanalysis outputs
over this remote region with low density of assimilated observations (Laloyaux et al., 2018).
**4.2 Shift of the annual cycle maximum**
For the end of the 21st century, the five selected ESMs project a delayed maximum near the poles and an earlier maximum in
many tropical continental regions (Fig. 6). The shift of the maximum between the two historical periods does not show such
distinct pattern (Fig. 5). In Europe, the southern and eastern regions experienced a delay of maximum of up to 10 days,



wherease northern Europe a slightly earlier maximum. In North America, a similar dipole pattern is seen between eastern and
western regions (Fig. 5).
The largest differences between the reanalyses and ESMs are found over southern America and eastern and southern Africa
(Fig. 5). Also, in the land regions near the equator, there is a disagreement between the two reanalyses (e.g., WSAF, MDG,
ESAF in Africa, and NWS and SAM in South America). This is mainly due to the fact that the annual cycle has no distinct
maximum peak and the warm season part of the annual cycle is rather flat; thus, a small temperature change in this season may
result in a large shift of the maximum (Fig. S03). In North-Central America (NCA), even though it is farther from the equator,
CERA20C gives a large shift of the maximum, but the actual temperature change is small, similar to regions NWS and SAM
in South America, which are closer to the equator. In most of the regions further from the equator, the reanalyses agree on the
sign of the shift in the maximum. In the oceanic regions near the equator, the reanalyses show a shift to an earlier onset of the
maximum. Both between the two historical periods and between the reference and future period, we see a smaller shift of the
maximum in Antarctica than in the Arctic (Fig. 5, 6).
In the regions near the equator, there are two distinct maxima of the annual cycle (Fig. S03, S04); therefore, we evaluate the
shift also for the second maximum (Fig. 5). We stress that the first/second refers to the earlier/later occurrence during the year,
not to the magnitude. In some of these regions, the annual cycle has even more "maxima"; it is modulated by at least three
peaks (Fig. S03, e.g., CNRM-ESM2 in north-eastern Africa (NEAF)). As the amplitude of the annual cycle is generally low
in near-equator regions, the whole curves are rather flat, and it is difficult to compare them between the datasets. For example,
in the oceanic part of south-eastern Asia (SEA region), the CERA20C reanalysis shows a large shift in the 2nd maximum (Fig.
5). However, in Fig. S03 it is clear that in the first historical period, the annual cycle near the 2nd maximum is very flat, and
therefore the large shift rather indicates a clearer emergence of the 2nd maximum. Also, in north-eastern Africa (NEAF), the
evaluation of the maximum shift is rather problematic. The maxima in different ESMs and reanalyses are shifted, so it is
actually questionable to compare them (Fig. S03). Similarly, in north-west southern America (NWS), the mean annual cycle
in the historical periods has, according to the reanalyses, only one distinct maximum (Fig. S03). However, the ESMs show a
second maximum. We do not consider it in our analysis, but it is interesting to note that the annual cycle in this region is
projected to change in the way that the temperature at this second maximum, not present in reanalyses, becomes higher than
the first maximum (Fig. S04). As the annual cycle in the near-equator regions is closely related to the seasonal distribution of
precipitation, the shift of the maximum can indicate the change in the occurrence of dry and wet seasons. Over Africa, the first
maximum is projected to occur earlier, and the second maximum is expected to be delayed.  We note that in above mentioned
regions with rather flat maximum and low amplitude, the ESMs and reanalyses mostly disagree on changes in amplitude (see
Section 4.4).
**4.3 Annual cycle velocity**
Higher latitudes detect a higher magnitude of temperature velocity change than lower latitudes. For future changes, the
geographical pattern of projected temperature velocity change remains the same as between the historical periods, with the



magnitude of change twice as high in most of the regions (Fig. 8). The velocity change over the oceans is mostly smaller
compared to the continents (Fig. 7, 8). Between the two historical periods, all regions experienced both a decrease and an
increase in the slope of the annual cycle, depending on the time of year (see Fig. 7 b, c and Fig. S05). Recent changes in
temperature velocity are largest in the western-central part of Euroasia (EEU, WSB, and ESB; Fig. 7). Generally, the regions
with larger changes in velocity have a larger range between the 10th and 90th percentiles, which is expected given the definition
of the diagnostic.
The temperature velocity changes agree between the reanalyses, except for Antarctica and the RFE (eastern Asia), and CNA
(central North America) regions. Further, the ESMs tend to underestimate the reanalysis-based velocity changes in the middle
and higher latitudes of the northern hemisphere, and largely agree on the smaller changes in the tropics and over the southern
hemisphere. The temperature velocity changes between the two historical periods are mostly in the interval between -0.1 and
+0.1 °C/day. This change of slope of the annual cycle curve can result in a temperature change of 3 °C per month. It naturally
corresponds to changes in the amplitude of the annual cycle and changes in the temperature contrasts between seasons.
However, Fig. S05 shows that the changes in the velocity are, in most cases, rather variable during the year, with the sign
persisting not for the whole season, but rather for a week up to two months. Still, except for a couple of regions, the two
reanalyses have rather similar annual cycle of temperature velocity changes in the historical periods. Unlike the temperature
change, the reanalyses agree on the sign and value of annual cycle velocity change over Greenland.
The five ESMs mostly follow the reanalysis-based pattern of change in temperature velocity. If there is any disagreement, the
models tend to underestimate the magnitude of changes. This is mainly seen in the northern hemisphere's higher latitudes.
Generally, over the northern hemisphere continents, we mostly see higher fluctuation of velocity changes between negative
and positive values in winter than in summer. Regarding the projections, Fig. S06 depicts a distinct annual cycle of velocity
changes in the regions where the expected warming is larger in one of the seasons. A nice example is the Arctic, where we can
see a decrease in velocity in the spring and the autumn, but near-zero or positive changes in winter and summer (Fig. S06).
This stems from a flattening of the annual cycle and higher warming in winter than in summer. In northern mid-latitude regions,
the velocity changes are more variable during winter than in summer (Fig. S06), which is connected to a  higher increase of
temperature in winter than in summer and shrinking amplitude, as discussed below.

### 4.4. Annual cycle amplitude

Both reanalyses agree on the prevailing decrease in amplitude, with the largest changes detected in EEU (eastern Europe) and
WSB (western Siberia), CNA, NWN (both northern America), the Arctic ocean, and Antarctica (Fig. 9). Unlike the temperature
change, both reanalyses show that the amplitude change in the Arctic is larger than in Antarctica. The ESMs, in turn, show
diverging changes of amplitude between the historical periods over the globe. Over the oceans, we see small amplitude changes
with varying signs in both models and reanalyses, except for the Southern Ocean and the Arctic region, where all the models
and reanalyses show an amplitude decrease (ERA5 -0.4 and CERA20C -1.1 °C in SOO, ERA5 -2.9, CERA20C -1.6 °C).
Clearly, decreasing annual cycle amplitude arises from a faster increase of temperature in the colder part of the year, in



comparison to the warm season (Fig. S07). Such seasonal difference in temperature trends during the 20th century has been
reported by Nigam et al. (2017) for the northern hemisphere. However, there are several regions where the reanalyses show
an increase in amplitude, up to 0.8 °C, e.g., Greenland, the southern part of South America, Madagascar, and interestingly also
Siberia (RAR) and some regions in northern America (Fig. 9). The ESMs show decreasing amplitude everywhere. We
hypothesize that the discrepancy between ESMs and reanalyses over Greenland could be connected to differences in the
evolution of sea ice between simulations. A recent increase in amplitude over Greenland has also been reported by Deng and
Fu (2023).
In many regions, the projected future amplitude changes have the sign opposite to the changes between the historical periods
(compare Fig. 9 and 10). The largest increase is projected over the Mediterranean and West-Central Asia regions (due to
summer warming being more than winter), and the largest decrease over the Arctic Ocean (due to winter warming being higher
in winter than summer). Generally, the amplitude increase is projected for most of the southern hemisphere and equatorial
areas, whereas most of the middle to higher latitudes of the northern hemisphere are projected to experience a decrease in the
amplitude. There are a few exceptions: West-Central Europe (WCE), East-Central Asia (ECA), and the western part of the
USA (WNA), where we can see an increase in the amplitude of app. 2 - 3 °C. The increasing amplitude indicates an increase
in thermal continentality of climate, with higher contrasts between winter and summer.
To illustrate how the individual FDA diagnostics complement each other, the projected amplitude changes correspond very
well to the projected temperature velocity changes; a higher increase in velocity is connected to a higher decrease in amplitude,
and the other way around.

## 5. Discussion

The shape of the mean temperature annual cycle can be considered a very basic feature of climate. Nonetheless, we highlight
large observational uncertainty related to its recent changes, i.e., distinct differences between the two reanalyses, especially
over equatorial and polar regions. Multiple differences between the reanalyses might be behind the discrepancies. Besides
differences in spatial resolution, CERA20C is a coupled reanalysis, whereas ERA5 was produced by the atmospheric model
only. Laloyaux et al. (2018) emphasize that the former is expected to be more realistic in terms of ocean heat balance and
ocean heat uptake, important for the temporal evolution of near-surface air temperature and its low-frequency variability.
Regarding the discrepancies over Antarctica, the assimilated observations are scarce and might be spurious (Laloyaoux et al.,
2018). Furthermore, as pointed out by, e.g., McKinon et al. (2024), the reanalysis performance is in general questionable over
regions that have spurious observations, not only in Antarctica but also over large portions of Africa or Southern America.
Furthermore, Yettella and England (2018) emphasized large internal climate variability uncertainty connected to the evolution
of annual cycle shape over northern hemisphere middle and high latitudes.
Even though we analyze only five CMIP6 ESMs, which is admittedly a very small subset of the whole multi-model ensemble,
they differ in many aspects, including spatial resolution (Table 2), model family (Merrifield et al., 2023), and climate sensitivity



(Table 2, Meehl et al., 2020). It is not thus surprising that they show diverse outcomes. They are not always able to reproduce
the reanalysis-based historical changes, and their projections differ in many aspects. The differences in the structure of the
models imply different character of internal climate variability, which is certainly behind some of the discrepancies. ESMs
with higher climate sensitivity generally project larger annual cycle shape changes (e.g., CanESM5 in the Arctic, Fig. 3, 4).
Even though it has been argued that the higher sensitivities are not plausible (e.g., IPCC, 2021), it is difficult to rule out the
hot models, especially in the case of regional impact assessment (Palmer et al., 2023; Swaminathan et al., 2024).
Previous studies on changes in the annual cycle mostly concentrated on the amplitude and shift of the maximum. Chen et al.
(2019) studied ERA-Interim-based and CMIP5-simulated spatial patterns of seasonal amplitude and phase. They concluded
that the seasonal amplitude reduced during the 21st century over high latitudes of both hemispheres because cold-season air
temperature increases faster than warm-season air temperature. In contrast, over low latitudes, the expected evolution is exactly
the reverse. Further, the maximum of the annual cycle was projected to be delayed by 15 - 30 days over the high-latitude
oceans where the sea ice is expected to shrink significantly (Chen et al., 2019). All these patterns are also obvious in our
results, implying consistency between CMIP5 and CMIP6 projections. The gradual decrease of amplitude prevailing over most
of the northern hemisphere has also been reported in other studies, including Stine and Huybers (2012), Wang and Dillon
(2014), Nigam et al. (2017), and Cornes et al. (2018). However, we depict some regions where the reanalyses disagree on the
sign of change, and also regions where both ESMs and reanalyses imply an increase in the amplitude. Delayed onset of annual
cycle maximum over most of the northern-hemisphere continents was also reported by Deng and Fu (2023).
In a recent study, Brunner and Voigt (2024) revealed a systematic bias in the definition of percentile-based temperature
extremes (Tx90p) when using too long seasonally running windows. One of the pitfalls they revealed was spurious signals of
change in Tx90p, as the strength of the bias depends on the shape of the temperature annual cycle (as well as the day-to-day
variability). They find two particularly affected regions: a region of increasing bias in oceans north of 45°N, except the very
highest latitudes (approximately our NPO, NAO, and MED regions), and a region of decreasing bias in our ARS region (see
their Fig. 5a). Connecting to our results, stronger seasonal gradients (corresponding to a higher temperature velocity) favour a
stronger bias in Tx90p (Brunner and Voigt 2024). Indeed, we find a weak (in particular compared to some land regions; see
Fig. 8), but clear increase in temperature velocity between 1961-1990 and 2071-2100 in all three regions affected by increasing
bias (NPO, NAO, MED), which is particularly pronounced towards and away from the annual maximum (which is located
around end of August/beginning of September or day of the year 240, Fig. S04). While the absolute value of the temperature
velocity change is considerably higher in other regions, its systematic increase in combination with the low day-to-day
temperature variability considerably contributes to the increase in Tx90p bias in these regions.
For the region of decreasing bias in Brunner and Voigt (2024), roughly corresponding to our ARS region, the attribution of
the bias change to the temperature velocity is less clear due to a combination of two reasons; first, the decreased bias stems
mainly from a very limited number of days surrounding the second annual minimum in the region (July and September; see
Fig. 5c in Brunner and Voigt 2024). For CanESM5, which was used in Brunner and Voigt (2024), we do find a short consistent
decrease in the temperature velocity corresponding to those months (Fig. S06). Second, the decrease in bias found in this



region is also (at least partly) attributable to an increase in day-to-day variability, which is not evaluated by the FDA diagnostics.

**Conclusions**

This paper presented an innovative method for assessing the shape of the annual cycle. It is applicable for different climatic variables and for various purposes, with no need to make any prior assumptions about the annual cycle shape. The diagnostics we introduced provide important information about different aspects of the seasonal cycle shape and its changes: amplitude, slope, and location of extrema. We analyze annual cycles averaged over 30-year periods. However, the method can also serve to analyze shorter-term variability of the seasonal cycle and even study inter-annual variability of the shape features. Unlike methods based on monthly or seasonal means (e.g., evaluating the amplitude based on monthly values), the FDA diagnostics can capture even slight changes in the shape of the annual cycle, for example, in the timing of the maximum, discussed in Section 4.2.

We have illustrated the methodology by using the example of the annual temperature cycle and its changes in pre-defined climatological regions. We used it to assess recent and projected changes in the annual temperature cycle in a selection of ESMs and observation-based datasets. Other potential applications include assessing other variables or evaluating the model performance explicitly. Differences between models and one or more reference datasets would be investigated in the latter case. The results can be aggregated, assessing the ensemble mean and spread. The diagnostics can be modified to evaluate not only changes between time periods, but also differences between datasets to reveal model biases in the representation of the annual cycle compared to an observational reference. The definition of FDA diagnostics can thus be tailored for specific interests and applications.

**Supplement:**

Additional figures S01-S08.

**Interactive computing environment**

Jupyter notebooks are published in Zenodo: DOI: 10.5281/zenodo.15866118

**Code availability**

Jupyter notebooks are published in Zenodo: DOI: 10.5281/zenodo.15866118



**Data availability**

Underlying CMIP6 data were downloaded from the ETH Zurich CMIP6 next generation archive (Brunner et al., 2020), but are also freely available in the ESGF. The underlying ERA5 data are freely available from the Copernicus Climate Data Store. Data from CERA20C reanalysis were downloaded from ECMWF with the help of Jan Masek from the Czech Hydrometeorological Institute. Preprocessed data used for the FDA calculations are published in Zenodo: DOI: 10.5281/zenodo.15866118. For any details or requests, please contact one the corresponding authors.

**Author contribution**

EH: Conceptualization, data pre-processing, interpretation of the results, writing of the draft. LB: Conceptualization, data pre-processing, plotting, interpretation of the results. JK: Methodology development, FDA calculations, coding, and plotting. All three authors have contributed to the writing of the final text.

**Competing interests**

The authors declare no competing interests.

**Acknowledgements**

We acknowledge the CMIP community for providing the climate model data, retained and globally distributed in the framework of the ESGF.

We acknowledge the European Centre for Medium-Range Weather Forecasts (ECMWF) for producing the ERA5 (Hersbach et al., 2020) and CERA20C (Laloyaux et al., 2018) reanalyses.

We acknowledge the World Climate Research Programme's Working Group on Coupled Modelling, which is responsible for CMIP, and we thank the climate modeling groups for producing and making available their model outputs.

For CMIP, the US Department of Energy's Program for Climate Model Diagnosis and Intercomparison provides coordinating support and led the development of software infrastructure in partnership with the Global Organization for Earth System Science Portals.

We thank the Copernicus Climate Change Service, ECMWF, and the ETH Zurich CMIP6 next generation archive (Brunner et al., 2020).

Data from CERA20C reanalysis were downloaded from ECMWF with the help of Jan Masek from the Czech Hydrometeorological Institute.



**Financial support**
This research was partly supported by the program of the Charles University Cooperatio "Sci-Physics". LB has received
funding by the Deutsche Forschungsgemeinschaft (DFG, German Research Foundation) under Germany's Excellence Strategy
EXC 2037 'CLICCS—Climate, Climatic Change, and Society' - Project No. 390683824, a contribution to the Center for Earth
System Research and Sustainability (CEN) of the University of Hamburg. EH and JK were supported by the grant GA25-
15855S of the Czech Science Foundation.

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



**Table 1:** Basic information about the two reanalysis datasets used in the present study.

| Acronym | Modeling center | Horizontal resolution of the atmospheric component (lat x lon) | Model components |
|---|---|---|---|
| ERA5 | European Centre for Medium-Range Weather Forecasts (ECMWF) | 0.28° x 0.28° (31 km x 31 km) | Atmosphere |
| CERA20C | European Centre for Medium-Range Weather Forecasts (ECMWF) | 1.125° x 1.125° | Atmosphere, Land, Ocean, Waves, Sea ice |






**Table 2:** Basic information on the CMIP6 ESMs used in the present study. The values of the equilibrium climate sensitivity are taken from
Meehl et al. (2020).

| ESM Acronym | Modelling center | Horizontal resolution (lat x lon) | Equilibrium climate sensitivity |
|---|---|---|---|
| CanESM5 | Canadian Centre for Climate Modelling and Analysis, Environment and Climate Change Canada, Victoria, Canada | 2.8° x 2.8° | 5.6 °C |
| CNRM-ESM2-1 | Centre National de Recherches Meteorologiques (CNRM) and Centre Europeen de Recherche et de Formation Avancee en Calcul Scientifique (CERFACS), Toulouse, France | 1.4° x 1.4° | 4.8 °C |
| EC-Earth3 | EC-Earth consortium, Rossby Center, Swedish Meteorological and Hydrological Institute/SMHI, Norrkoping, Sweden | 0.7° x 0.7° | 4.3 °C |
| MPI-ESM1-2-HR | Max Planck Institute for Meteorology, Germany | 0.94° × 0.94° | 3.0 °C |
| NorESM2-MM | NorESM Climate modeling Consortium, Oslo, Norway | 1.25° × 0.9° | 2.5 °C |




**Table 3:** Overview of time periods investigated in the present study.

| Time period | Notation | Datasets analyzed |
|---|---|---|
| 1951-1980 | Historical (first) | Reanalyses and the five selected ESMs |
| 1981-2010 | Historical (second) | Reanalyses and the five selected ESMs |
| 1961-1990 | Reference | Five selected ESMs and their multi-model mean |
| 2071-2100 | Future/scenario | Five selected ESMs and  their multi-model mean |






**Table 4:** Overview of the FDA diagnostics and the corresponding figures showing the results. The "Regional figures" in the Supplement
show the FDA results underlying the individual diagnostics.

| Diagnostic | Global figures (historical / projections) | Regional figures (historical / projections) |
|---|---|---|
| Annual cycle shape | Fig. 3 / Fig. 4 | Fig. S01 / Fig. S02 |
| Annual cycle maximum | Fig. 5 / Fig. 6 | Fig. S03 / Fig. S04 |
| Annual cycle velocity | Fig. 7 / Fig. 8 | Fig. S05 / Fig. S06 |
| Annual cycle amplitude | Fig. 9 / Fig. 10 | Fig. S07 / Fig. S08 |




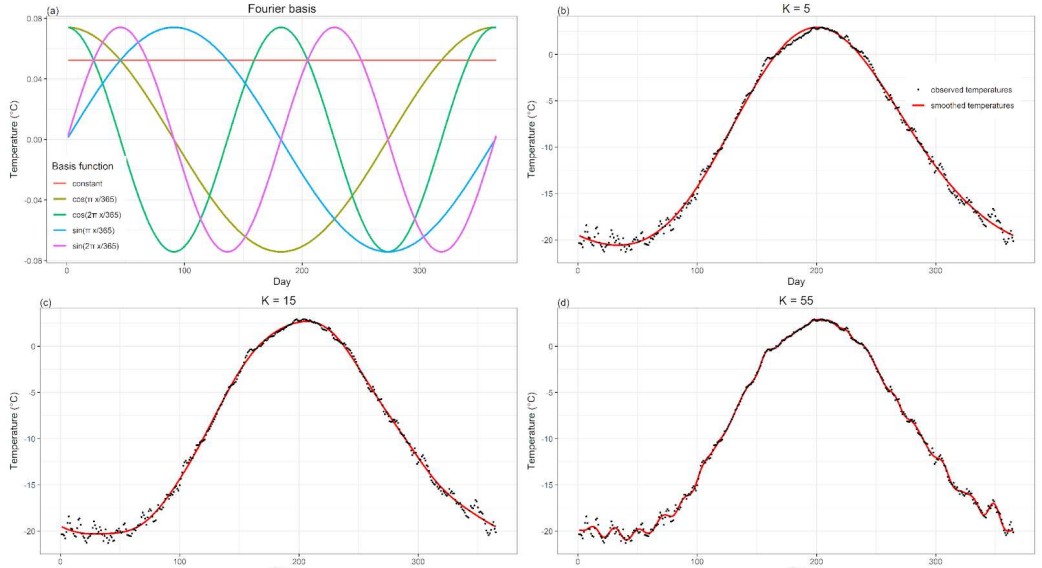

**Figure 1.** (a) Basis functions for the case K=5; (b, c, d) smoothed temperature with respect to K = 5, 15, 55. As the number of coefficients
grows more and more variability beyond the mean seasonal cycle is captured by the FDA.






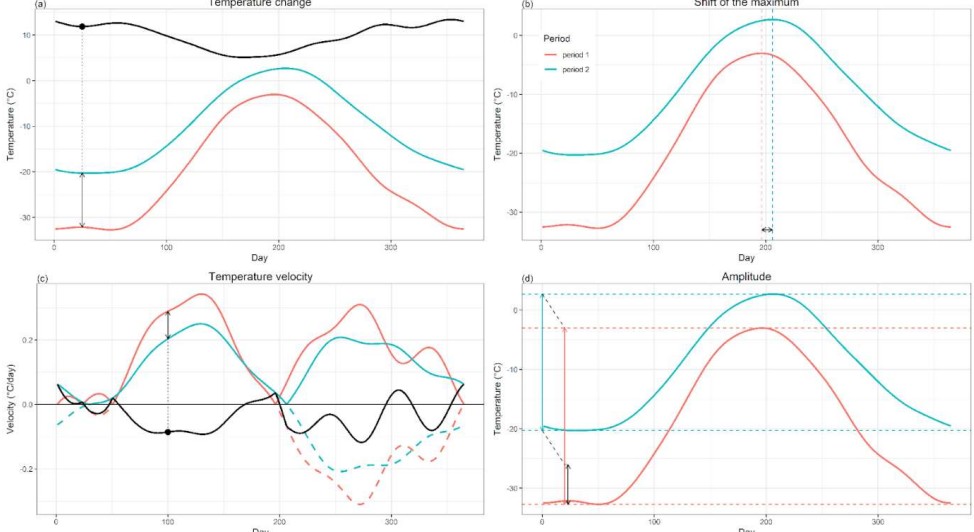

**Figure 2.** The FDA diagnostics interpretation framework. Blue and red lines illustrate an example FDA-smoothed temperature annual cycle in two time periods, except for panel (c), where the lines represent the absolute values of the 1st derivative of the FDA-smoothed curves, i.e., temperature velocity, and the dashed color lines are used to depict negative temperature velocities. (a) The black arrow corresponds to the vertical temperature change between the two periods on a specific day of the year, and the black line represents its values during the whole year. (b) Dashed lines represent the days of temperature maxima, and the black arrow corresponds to the shift of the maximum. (c) The black arrow corresponds to the change of temperature velocity between the two periods on a specific day, and the black line represents its values during the whole year. (d) The blue and red arrows correspond to the amplitudes in each period, and the vertical black arrow illustrates the change in the amplitude between the two periods.

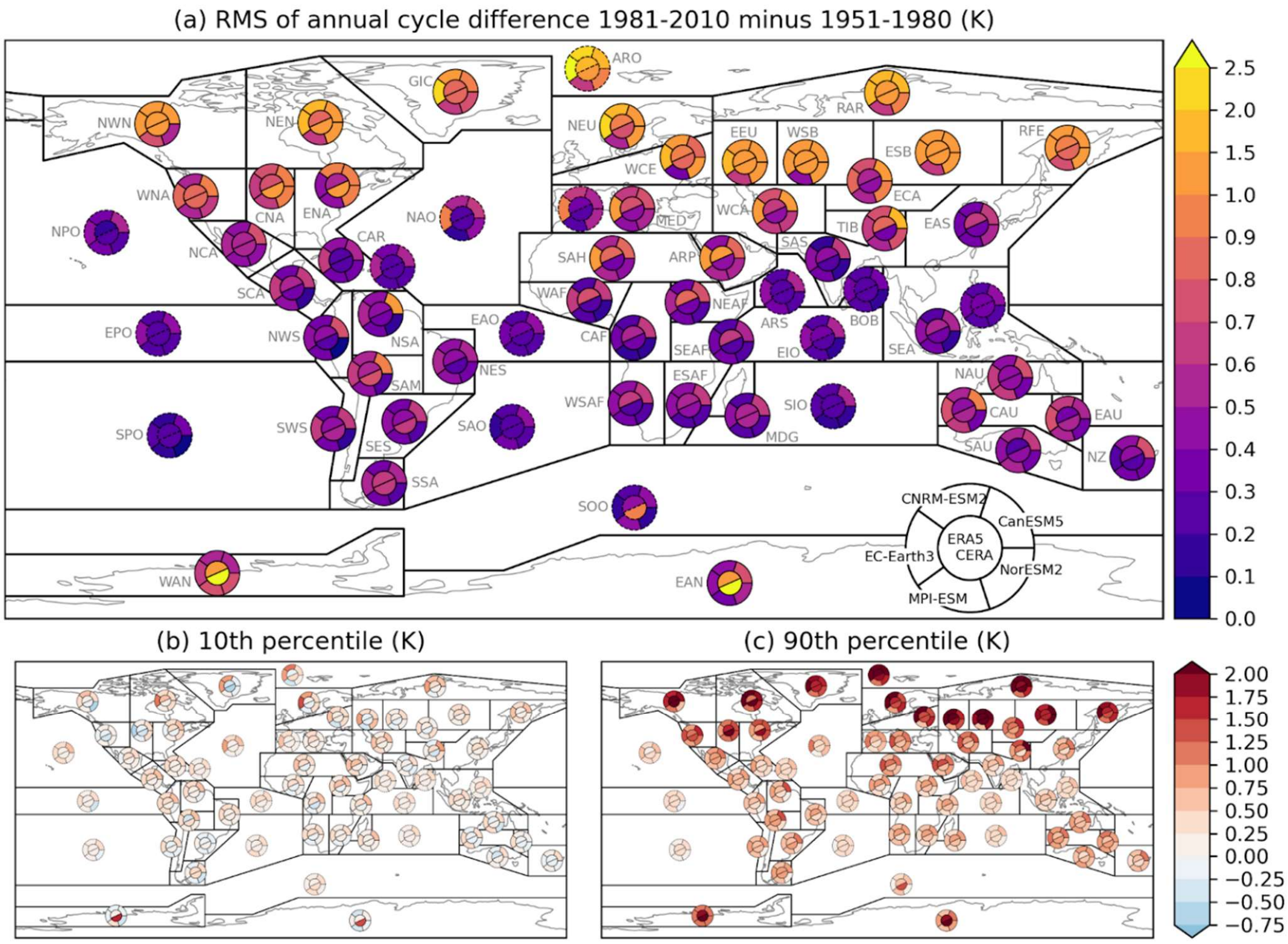

**Figure 3:** (a) Root mean square difference (Euclidean distance, [K]) of the whole FDA-smoothed mean annual cycle curves between the two historical periods 1981-2010 and 1951-1980. (b) 10th percentile and (c) 90th percentile of daily distances [K] between the smoothed annual cycle curves. For each region, the center of the pie plot shows results based on the two reanalysis datasets ERA5 and CERA, while the outer part of the pie shows the results for the five CMIP6 ESMs (see Section 2 for data description).




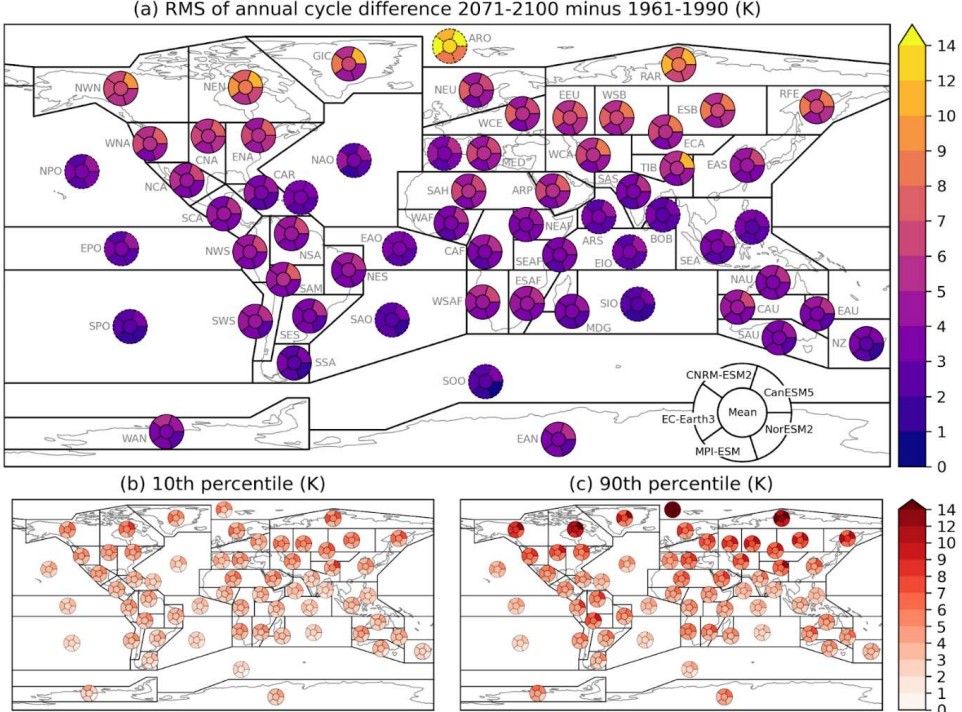

**Figure 4:** Same as Fig. 3, but for the differences between the scenario period 2071-2100 and the reference period 1961-1990. Future model
simulations follow the SSP3-7.0 socio-economic pathway.







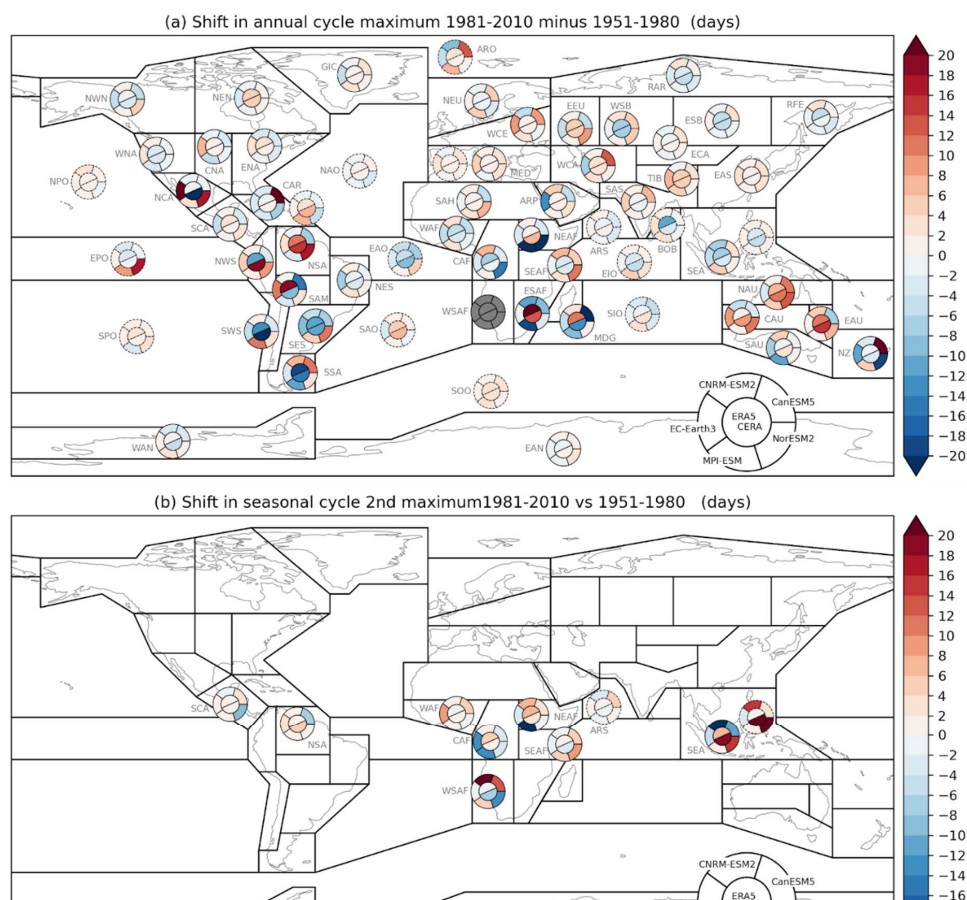

**Figure 5:** Similar to Fig. 3a, but for (a) shift in the annual cycle maximum and (b) shift of the second maximum in regions with two distinct maxima. Note that the "first" and "second" maxima are considered chronologically from January 1st, with no regard to the actual maximum magnitude (the second maximum can potentially have a higher temperature than the first).





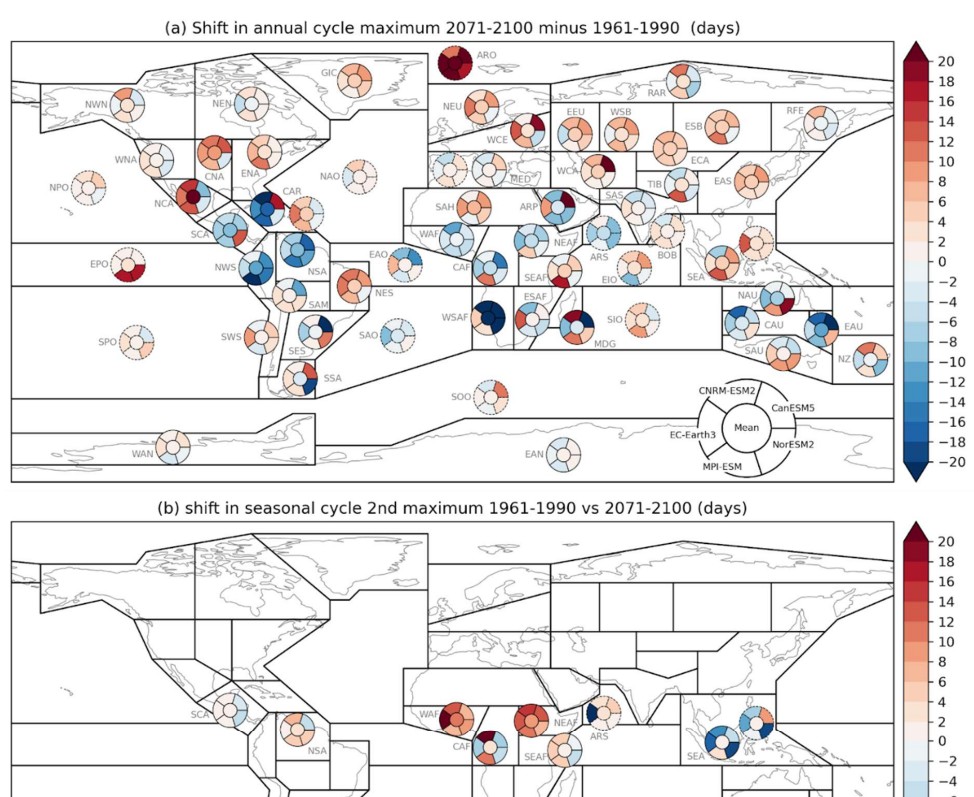


**Figure 6:** Same as Fig. 5, but for the shifts of the maxima between the scenario and historical periods.




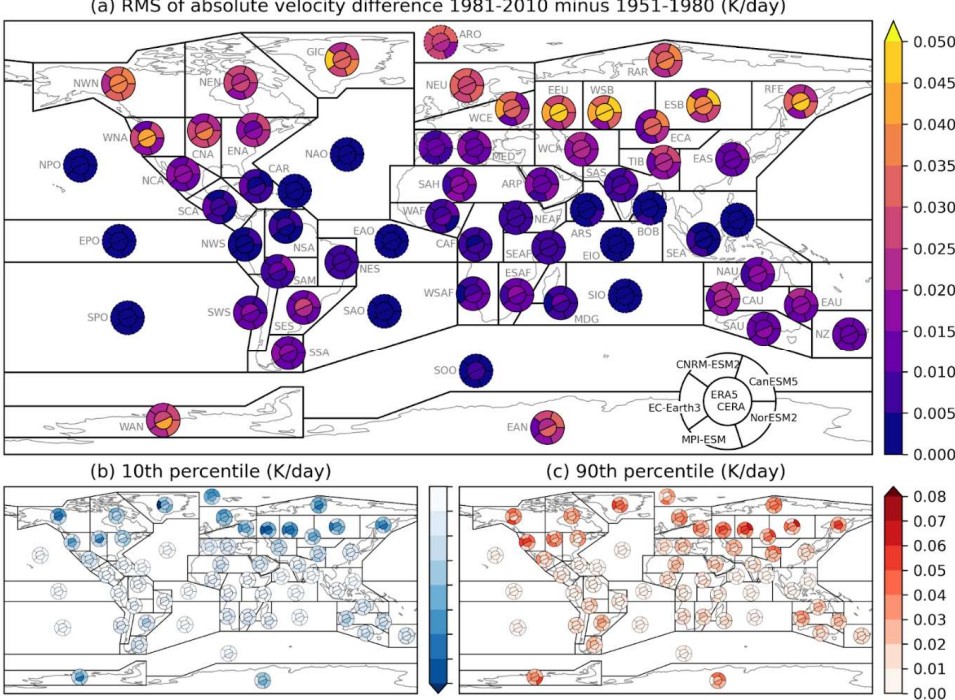

**Figure 7:** Same as Fig. 3, but for temperature velocity, that is the 1st derivative of the smoothed curve of the temperature annual cycle. Note that the color scale for plot in (b) has the same range as plot (c), just reversed, i.e., negative values going from 0 K/day (white color) to -0.08 K/day (darkest blue). The range is not shown for the sake of better visibility of the plots.



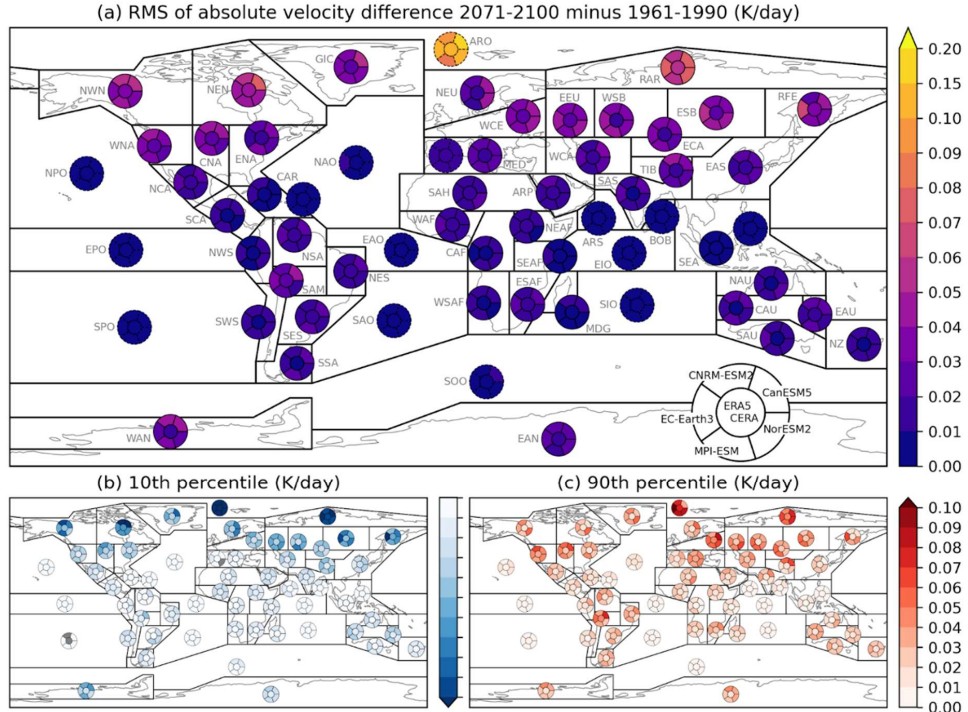

**Figure 8:** Same as Fig. 7, but for the change in temperature velocity between the scenario and reference periods.



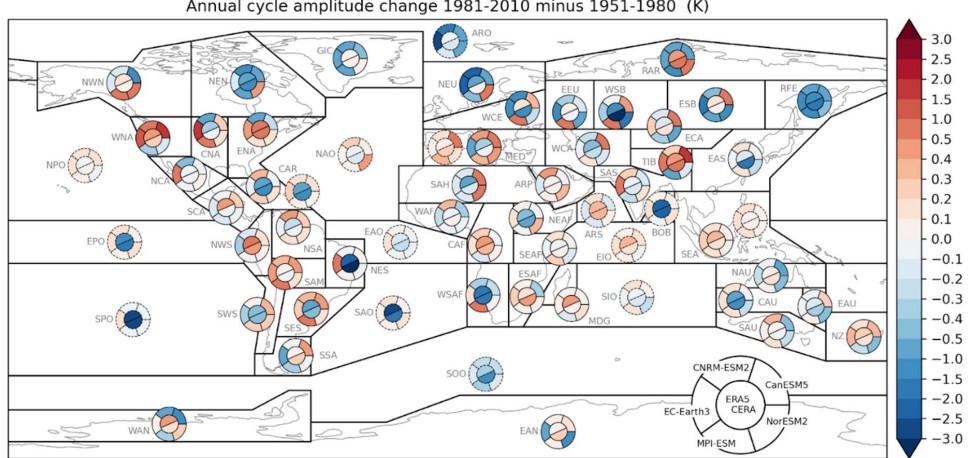

**Figure 9:** Same as Fig. 3a, but for the change in the amplitude of the annual cycle.



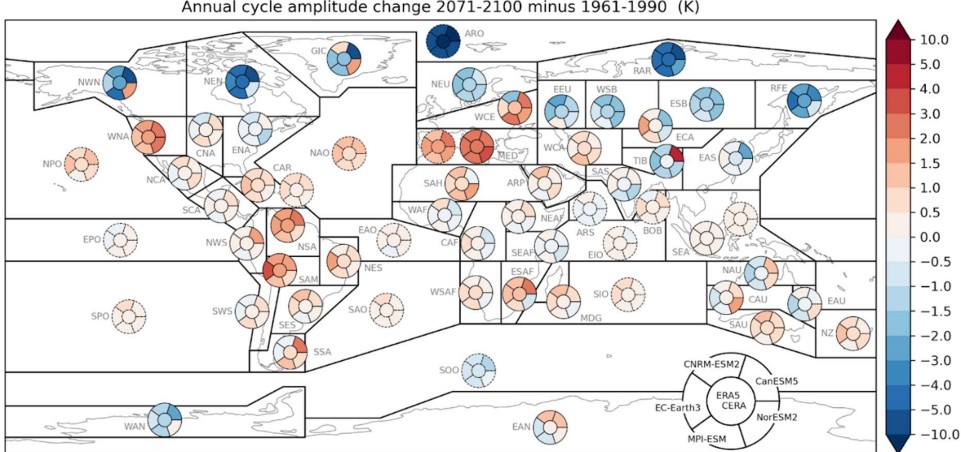

**Figure 10:** Same as Fig. 4a, but for the change in the amplitude of the annual cycle.