# Peer review of "Quantifying changes in seasonal temperature variations using a functional data analysis approach"

_EGUsphere, 2025_

## Referee Comment (RC2)

**Review for Earth System Dynamics egusphere-2025-3360**

**Title:** "Quantifying changes in seasonal temperature variations using a functional data analysis approach"

**Authors:** Eva Holtanová, Jan Koláček, Lukas Brunner

**General Comments**

The study offers a novel approach to study the annual daily cycle of temperature over the globe by means of a FDA (functional data analysis) applied to reanalysis (CERA20C and ERA5) and an ensemble of CMIP6 ESMs for the whole XXIst century. It looks quite interesting in terms of a better description of this very relevant feature to characterize the climate of each region of the globe. The comparison among reanalysis and global models, differences of regions, and how this is being modified when looking at different periods, and so how anthropogenic climate change is expected to modify temperature in terms of its annual cycle. Therefore, it looks that it could be considered by the journal for publication, once the different comments and suggestions were considered by the authors.

**Specific Comments**

1. An accurate and direct title is always welcomed, but, at the same time, as precise as possible should also be important. I miss some reference to "annual cycle features" and time periods and spatial coverage (global, present/future), for a first understanding of what the work is about.

2. I miss several references to previous studies, apart from the ones mentioned on the introduction: For example, apart from the Lopez-Franca et al., 2022, mentioned on line 45, also Lopez-Franca et al., 2013: "Changes in the onset and length of seasons from an ensemble of regional climate models over Spain for future climate conditions", DOI: 10.1007/s00704-013-0868-2, although only focused on one region, seems to be of interest, based on temperature (maximum and minimum). More references can also be found when searching "thermal seasons" expression: Ruosteenoja K, Markkanen T, Räisänen J. "Thermal seasons in northern Europe in projected future climate". Int J Climatol. 2020; 40: 4444–4462. https://doi.org/10.1002/joc.6466; Tu, K., Yan, Z., & Qian, C. (2024). Understanding seasonal cycle of daily extreme temperatures based on generalized additive model for location, scale and shape with smoothing spline. Int. J. Climatol., 44(6), 1883–1897. https://doi.org/10.1002/joc.8430 or Hekmatzadeh, A.A., Kaboli, S. & Torabi Haghighi, A. "New indices for assessing changes in seasons and in timing characteristics of air temperatur". Theor Appl Climatol 140, 1247–1261 (2020). https://doi.org/10.1007/s00704-020-03156-w; Liu, F., Song, F., & Luo, Y. (2024). "Human-induced intensified seasonal cycle of sea surface temperature". Nature Comms, 15(1), 3948, among others, could be mentioned. I know it is hard to make a complete overview of bibliography, but I miss some more references for example when talking about summer lengthening (line 55), such as Peña-Ortiz, C., Barriopedro, D., & García-Herrera, R. (2015). Multidecadal variability of the summer length in Europe. J. Climate, 28(13), 5375-5388, for example.

3. FDA is first pointed on the abstract, but for a non-expert reader, perhaps a more intuitive description of it could be made there?.

4. I have concern about the precise definition of "absolute temperature", first on line 21. What does it exactly mean?. Mean daily temperature in K?

5. More widely talking about the variable used on the work, did the authors made a thought about applying it to other temperature variables, such as daily maximum, minimum or daily range of temperature?. Could it potentially add interest to the proposed analysis or methodology?. Something is said on conclusions (line 363), but perhaps some comments could be made on methods or introduction, or at least on an answer to this question here.

6. When talking about temperature cycle changing on the recent decades (lines 52-63), which of then are based on observations, reanalysis or models?

7. Regional features are mentioned on line 74. What about using also regional climate models on a study like the one presented here?. I understand that it is not possible to study all the elements of climate, but probably it could be indicated as one of the several lines of further analysis, I guess.

8. On line 89 the data used in the study is presented, and so it is stated that "the mean annual cycle of near-surface air temperature" is going to be used. What about being more precise, indicating that mean daily data is used?.

9. On line 153 when starting the analysis indices with the annual cycle shape, annual extremes are defined in some way from $10^{th}$ and $90^{th}$ percentiles. Did the authors tested more extreme percentiles (95-5) as more extreme representation of changes extreme conditions?.

10. Going back to the potential usage of maximum and minimum temperature, to further inspect annual cycle of temperature, as from a purely physical perspective, at least max and min temperatures are a good representation, much more than mean daily values to processes of the climatic system that determine both values, related to local or synoptic mechanisms. I do not mean to add those variables to the study, but maybe the authors could make a comment about this idea, at least on these answers.

11. The word "shape" is used for the first index, and it is measured in degrees. Maybe accumulated/integrated temperature change could better define what is obtained there?.

12. Related to the parameter named "shift" of the maximum, I understand the reason for obtaining it from the proposed methodology, but that date is strongly and mainly related to astronomical features and thermal inertia more than to climatic conditions. On the contrary, season length and start/end of them seems to be more interesting, as widely studied by previous works. Do the authors have a comment about this topic?.

13. On line 109 climatological regions are indicated, using the IPCC (Iturbide et al., 2020) definitions. It is clear that they are many times too big to distinguish real regional climates, although it is clear that it would be impossible to have more spatial detail without making the work too long. But would it have been possible to have, at least on the supplementary material some purely global map grid by grid of some of the indices for some period and all the reanalysis and models to see how they look like in terms of spatial distribution?.

14. Even with that spatial distribution of regions, and considering the discussion presented on the work, perhaps a little bit more of analysis could be made about land/ocean differences.

15. A couple of big concerns when reading the results section is about the structure, that I would like to hear from the authors. One is about the time periods, and the other is about the order of the presented results, both are connected on my question. Comparison of two observational/historical periods (1951-1980) vs (1981-2010), and then future vs present periods (2071-2100) vs (1961-1990) are studied. In both cases, trends or differences are studied, and somehow compared. But both historical periods, purely in terms of temperature are not expected to present large changes, on the contrary to future scenarios analysis, and sometimes comparison of differences between both(e.g., 2°C, line 186 for historical periods, 10°C for future against present in line 201) makes a little bit confusing the interpretation of changes. Of course, differences on the first comparison is of 30 years, and it is of 90 years on the second comparison, when climate change signal is much more important. Perhaps directly study historical period comparing reanalysis against models, and then climate change signal would have been a more clear approach?.

16. My other concern is, related to this point, that if would have been a better results structure to first use the four indices for that historical period analysis (or the evaluation commented just before), and then the same for the climate change signal?. How do the authors would feel about that view?.

17. Several times, changes of the indices could be roughly related to global temperature trends or greenhouse gases rhythm of changes during the analysed periods. I guess if some idea about a potential relation with them could be added or commented during the analysis of the obtained results.

18. Some comments are made about the precipitation role on the annual cycle over some parts of the globe (line 244), and so the dry/wet seasons more than the warm/cold seasons over extratropical regions. Do the authors think that perhaps over some regions the annual cycle of temperature is not very relevant, and so their potential changes?.

19. One final remark: on line 358 it is said that many methods based on monthly variables, but most of the references studies seem to be using daily values, perhaps a more clear statement should be made when talking about other methods.

---

## Author Comment (AC1)

**Reply to the comments of Reviewer 1:**

This study uses a new methodology based on Functional Data Analysis (FDA) to analyze changes in the annual cycle of temperature for different regions of the globe. Changes are examined for the second half of the 20th century and projections into the end of the 21th century. Two reanalysis products (CERA20C and ERA5) and five CMIP6 earths system models are analyzed. Specific diagnostics of the annual cycle documented are the changes in absolute temperature, shifts in the maximum temperature, absolute velocity, and changes in the amplitude of the seasonal cycle.

The study is interesting and presents valuable diagnostics for understanding past and future changes in the annual cycle. I feel this paper is a valuable contribution to the literature. A few comments are listed below.

Line 193-194: Can you expand here on why there may be disagreement between the reanalyses in these situations, and if one may be more believable?

The sentence in question is: "Larger disagreement between the reanalyses also occurs over the southern ocean and in some regions near the equator (e.g., SAH and ARP)".

The discrepancies are caused by large observational uncertainty and large internal variability in southern high latitudes, in our opinion.

Due to the large observational uncertainty, it is really hard to assess which of the reanalyses should be considered more reliable, and it is beyond the scope of our study to hypothesize in this regard. Brunner et al. (2025) emphasize that the discrepancies between different reanalyses are larger in the southern ocean than in other ocean basins, not only for ERA5 and CERA, but also for other 8 reanalyses. Casado et al. (2023) even mention the possibility that the polar amplification, so far generally considered larger in the northern hemisphere than in the southern hemisphere, might be (or become in the near future) more pronounced in the southern hemisphere.

Figure: Width of the 20-year (1980-1999) temperature climatology from 10 observation-based datasets calculated as the maximum minus the minimum value at each grid cell. Note the different step sizes for each of the three shadings, used to roughly highlight the three regimes of uncertainty (ocean, land, high latitudes). The 10 datasets used are: 20CR, Berkeley Earth, ERA40, ERA-Interim, ERA5, JRA55, JRA3Q, MERRA, MERRA2, and NCAR-NCEP. Figure reprinted from Brunner et al. (2025).

Recent studies (e.g., Xin et al., 2023; Turner et al., 2020) described that during the second half of the 20th century, there was warming in the west and slight cooling in the east. This feature probably reversed after 2000, but we do not see any difference between the regions in terms of warming. The lack of this feature in our results is probably related to the Southern Annual Mode. Interestingly, we do not see any difference in the warming rates between West Antarctica (which includes the Antarctic Peninsula) and Eastern Antarctica. Maybe in our case, it is masked by the long-term averaging, but it might also be the result of uncertainty in both used reanalyses.

Moreover, the big difference between CERA and ERA5 in southern high latitudes, unlike in northern high latitudes, points to the importance of the coupling between the atmosphere ocean for the southern high latitudes. As demonstrated by, e.g., Kang et al. (2023), there is a strong relationship between tropical and subtropical Pacific and temperature changes in the southern ocean, and the simulation of these features is expected to be different in ERA5 (atmosphere only) than CERA (coupled simulation). The coupling does not automatically guarantee a better simulation, naturally.

We suggest that we will add the following text (a shorter version of the above discussion) into the first paragraph of the Discussion section, where we already discuss the observational uncertainty in high latitudes:

"Brunner et al. (2025) emphasize that the discrepancies between different reanalyses are rather larger in the southern ocean than in other ocean basins, not only for ERA5 and CERA20, but also for other 8 reanalyses. Moreover, the big difference between CERA20 and ERA5 in southern high latitudes, unlike in northern high latitudes, points to the importance of the coupling between the atmosphere ocean for the southern high latitudes. As demonstrated by, e.g., Kang et al. (2023), there is a strong relationship between tropical and subtropical Pacific and temperature changes in the southern ocean, and the simulation of these features is expected to be different in ERA5 (atmosphere only) than CERA (coupled simulation). The coupling does not automatically guarantee a better simulation, naturally." New references:

Brunner, L., Ghosh, R., Haimberger, L., Hohenegger, C., Putrasahan, D., Rackow, T., Knutti R., Voigt, A.: Three decades of simulating global temperatures with coupled global climate models, submitted to communications Earth & Environment, 2025.

Kang, S.M., Ceppi, P., Yu, Y. *et al.* Recent global climate feedback controlled by Southern Ocean cooling. *Nat. Geosci.* **16**, 775–780 (2023). https://doi.org/10.1038/s41561-023-01256-6

Turner, J., Marshall, G. J., Clem, K., Colwell, S., Phillips, T., & Lu, H. (2020). Antarctic temperature variability and change from station data. *International Journal of Climatology*, 40(6), 2986-3007.

Xin, M., Clem, K. R., Turner, J., Stammerjohn, S. E., Zhu, J., Cai, W., & Li, X. (2023). Westwarming East-cooling trend over Antarctica reversed since early 21st century driven by large-scale circulation variation. *Environmental Research Letters*, *18*(6), 064034.

Line 277-278: In Figure 9, it appears to me that the EEU and NWN have experienced an increase in amplitude. Please clarify.

We are sorry for this; we accidentally inserted an old version of the plot with erroneous results. We have updated the plot, see below, we will also update it in the revised version of the paper. The results in the supplementary plots are correct and show more details about the shift in temperature and the amplitude change.

Discussion: I think it should also be considered how the number of assimilated observations in the reanalysis products changes between the 2 observed periods (1951-1980 and 1981-2010), and what effect this may have on the results.

MPI-ESM

Thank you for pointing this out. We will add this comment into the revised text, into the first paragraph of the Discussion, when discussing the observational uncetainty:

"Moreover, the number of observations in both CERA20 and ERA5 were increasing during the study period, which might have influenced the results. In case of CERA, in which only variables measured over the ocean are assimilated, the data inputs from ships more than doubled, and data from buoys started to be assimilated after 1970 (Laloyaux et al., 2018). For ERA5, the number of assimilated observations increased from 53 000 to 570 000 between 1950 and 1970 (Bell et al., 2021)."

**new reference:**

Bell, B., Hersbach, H., Simmons, A., Berrisford, P., Dahlgren, P., Horányi, A., ... & Thépaut, J. N. (2021). The ERA5 global reanalysis: Preliminary extension to 1950. Quarterly Journal of the Royal Meteorological Society, 147(741), 4186-4227.

---

## Author Comment (AC2)

**Reply to the comments of Reviewer 2:**

An accurate and direct title is always welcomed, but, at the same time, as precise as possible should also be important. I miss some reference to "annual cycle features" and time periods and spatial coverage (global, present/future), for a first understanding of what the work is about.

Thank you to the reviewer for pointing this out. We agree and have updated the title to "A global perspective on past and future change in regional seasonal cycle shape".

Regarding the terminology, we added a note into the first paragraph of Section 3.2: "Regarding the terminology, we note that we use both terms "seasonal cycle" and "annual cycle" intermittently in the text, with no difference in its meaning."

2. I miss several references to previous studies, apart from the ones mentioned on the introduction:

For example, apart from the Lopez-Franca et al., 2022, mentioned on line 45, also Lopez-Franca et al., 2013: "Changes in the onset and length of seasons from an ensemble of regional climate models over Spain for future climate conditions", DOI:10.1007/s00704-013-0868-2, although only focused on one region, seems to be of interest, based on temperature (maximum and minimum). More references can also be found when searching "thermal seasons" expression:

Ruosteenoja K, Markkanen T,

R"ais"anen J. "Thermal seasons in northern Europe in projected future climate". Int J Climatol. 2020; 40: 4444–4462. https://doi.org/10.1002/joc.6466;

Tu, K., Yan, Z., & Qian, C. (2024). Understanding seasonal cycle of daily extreme temperatures based on generalized additive model for location, scale and shape with smoothing spline. Int. J. Climatol., 44(6), 1883–1897. https://doi.org/10.1002/joc.8430 or Hekmatzadeh, A.A., Kaboli, S. & Torabi Haghighi, A. "New indices for assessing changes in seasons and in timing characteristics of air temperatur". Theor Appl Climatol 140, 1247–1261 (2020). https://doi.org/10.1007/s00704-020-03156-w;

Liu, F., Song, F., & Luo, Y. (2024). "Human-induced intensified seasonal cycle of sea surface temperature". Nature Comms, 15(1), 3948,

among others, could be mentioned. I know it is hard to make a complete overview of bibliography, but I miss some more references for example when talking about summer lengthening (line 55), such as

Pena-Ortiz, C., Barriopedro, D., & Garc1a-Herrera, R. (2015). Multidecadal variability of the summer length in Europe. J. Climate, 28(13), 5375-5388, for example.

Thank you for pointing these references out. We will some citations into the revised version of the text:

in the Section 4.4: "Liu et al. (2024) showed an increase in the amplitude of SSTs over most of the ocean basins in recent 40 years. Our study period is longer, and the amplitude increase attributed to the anthropogenic forcings is probably masked to some extent by internal climate variability."

In the Introduction: References to Hekmatzadeh et al. (2020); Peña-Ortiz et al. (2015); A sentence into the Introduction: "Ruosteenoja et al. (2020) describe projected lengthening of the summer season in Northern Europe and López de la Franca et al. (2013) show the same for Spain, together with the winter season practically disappearing."

3. FDA is first pointed on the abstract, but for a non-expert reader, perhaps a more intuitive description of it could be made there?.

Thank you for pointing this out. The abstract sentence in question: "Here, we introduce an innovative approach based on Functional Data Analysis (FDA), a relatively new statistical approach."

We will modify this as follows: "We introduce the Functional Data Analysis (FDA) approach, representing the mean annual cycle by a linear combination of Fourier bases. The FDA methodology does not require any prior assumptions about the shape of the temperature seasonal cycle except periodicity and allows to quantitatively assess various aspects of the seasonal cycle shape."

- 4. I have concern about the precise definition of "absolute temperature", first on line 21. What does it exactly mean?. Mean daily temperature in K?

  Thank you for pointing this out. This is a typo we will correct the sentence: "We concentrate on diagnostics that evaluate the absolute change in temperature..."
- 5. More widely talking about the variable used on the work, did the authors made a thought about applying it to other temperature variables, such as daily maximum, minimum or daily range of temperature?. Could it potentially add interest to the proposed analysis or methodology?. Something is said on conclusions (line 363), but perhaps some comments could be made on methods or introduction, or at least on an answer to this question here. Thank you for mentioning this. We will extend the note about potential future directions of further studies in the conclusions (I. 363): "Other potential applications include assessing other variables, for example, minimum or maximum air temperature. The changes in the shape of annual cycle of these variables can have implications for the occurrence of extreme cold or heat events. Another possibility is to apply our method for explicitly evaluating model performance."
- 6. When talking about temperature cycle changing on the recent decades (lines 52-63), which of then are based on observations, reanalysis or models?

  To make this clear, we will modify the first sentence of the paragraph to "A large number of previous studies have shown that the **observed** shape of the temperature annual cycle has already changed in recent decades, …"

  For the papers in the second part of the paragraph, we already specify the relation to observations or models in the text.
- 7. Regional features are mentioned on line 74. What about using also regional climate models on a study like the one presented here?. I understand that it is not possible to study all the elements of climate, but probably it could be indicated as one of the several lines of further analysis, I guess.

Yes, thank you for this comment, we will extend the note about further use of the method in the conclusions (I. 363), saying that if regional models or ESMs in high resolution are used, we could identify more details about regional features. Specifically, we will add this text: "If the evaluation is applied to the outputs for regional climate models or ESMs with finer resolution, the assessment can be done for smaller geographical regions, revealing more details about projected changes and their potential impacts."

8. On line 89 the data used in the study is presented, and so it is stated that "the mean annual cycle of near-surface air temperature" is going to be used. What about being more precise, indicating that mean daily data is used?

To make this point clear, we will modify the sentence on I. 89 "The present study deals with the mean annual cycle of near-surface air temperature." to "The present study deals with the annual cycle of daily mean near-surface air temperature."

9. On line 153 when starting the analysis indices with the annual cycle shape, annual extremes are defined in some way from 10th and 90th percentiles. Did the authors tested more extreme percentiles (95-5) as more extreme representation of changes extreme conditions?.

Thank you to the reviewer for this question. We have selected 10th and 90th percentile to investigate the tails of the distribution of the daily temperature changes as they are frequently used thresholds (e.g., in the tempeature-based ETCCDI indices; Zhang et al. 2011, https://onlinelibrary.wiley.com/doi/abs/10.1002/wcc.147). In the supplementary figures, we show the days of the year that fall within the lowest and highest 10 percent of the values, i.e., the values lower/higher than 10th/90th percentile. Our focus in this study is not the extremes in particular, rather, we want to highlight cases when the annual mean change is positive, while there are still days that experience a negative change. The same is true for a moderate annual mean warming: it does not rule out considerably higher changes in some part of the year. While considering other percentiles might add some additional detail to this analysis (e.g., fully sample the seasonal deviations from the annual mean change), this is not the focus of the present study but could be interesting for future research.

10. Going back to the potential usage of maximum and minimum temperature, to further inspect annual cycle of temperature, as from a purely physical perspective, at least max and min temperatures are a good representation, much more than mean daily values to processes of the climatic system that determine both values, related to local or synoptic mechanisms. I do not mean to add those variables to the study, but maybe the authors could make a comment about this idea, at least on these answers.

Thank you for noting this. We have extended the comment on the potential further use of the method in reply to comment nr. 5.

- 11. The word "shape" is used for the first index, and it is measured in degrees. Maybe accumulated/integrated temperature change could better define what is obtained there?. Thank you for this suggestion. We have changed the term to "annualy integrated temperature change". The change resulted in some slight changes in the text and Table 4.
- 12. Related to the parameter named "shift" of the maximum, I understand the reason for obtaining it from the proposed methodology, but that date is strongly and mainly related to astronomical features and thermal inertia more than to climatic conditions. On the contrary, season length and start/end of them seem to be more interesting, as widely studied by previous works. Do the authors have a comment about this topic?.

We agree with the reviewer that the location of the maximum is indeed mainly determined by solar radiation and thermal inertia. However, the same is not necessarily true for *shifts* in the maximum, and a range of other factors might be relevant for this, including various feedbacks, e.g., related to soil moisture, snow cover over the continents, and sea ice over

high-latitude oceans. Near the equator, the occurrence of the intertropical convergence zone also plays a role in when the temperature maximum occurs and how sharp it is.

13. On line 109 climatological regions are indicated, using the IPCC (Iturbide et al., 2020) definitions. It is clear that they are many times too big to distinguish real regional climates, although it is clear that it would be impossible to have more spatial detail without making the work too long. But would it have been possible to have, at least on the supplementary material some purely global map grid by grid of some of the indices for some period and all the reanalysis and models to see how they look like in terms of spatial distribution?. This is an interesting idea. We decided to use the regions from Iturbide et al. (2020) as they are widely used and defined to represent relatively uniform climatic conditions. We consider these appropriate for a global to regional analysis.

Moreover, from our experience with the data used in the present study, when trying to create such a detailed map as suggested, we would encounter several issues. Mainly, ESMs and the reanalyses have all different resolutions and grid spacing; therefore, a remapping to a common grid would be necessary. The common grid would then be such that some models are closer to their native grid and others further away. Further, on a lat-lon grid, the area represented by each grid cell would change with latitude, affecting the influence of internal variability. Finally, the results would become noisy, especially near the equator, and issues with assigning the first and second maxima of the seasonal cycle would have to be solved. Therefore, we limit ourselves to the IPCC regions for this study.

14. Even with that spatial distribution of regions, and considering the discussion presented on the work, perhaps a little bit more of analysis could be made about land/ocean differences.

Thank you to the reviewer for this suggestion. We will add additional discussion on this to the text:

"It is generally expected and observed that the temperature changes over land would be higher than over the ocean (e.g., Sutton et al., 2007). Our results confirm this expectation in most of cases, especially in middle latitudes, when comparing regions for large ocean basins and surrounding regions (e.g., Fig. 3, 4), and when comparing the oceanic and land parts of the marginal sea regions (e.g., Mediterranean, Caribbean, south-east Asia). Near the equator, especially in the eastern Pacific Ocean (EPO) region, the position of the annual cycle maximum is expected to change more in the future than over continental regions in South America. This is also a region with relatively high disagreement between individual models. The uncertainty is apparently connected to rather flat maximum and low amplitude of the seasonal cycle, with even a small temperature change in a part of the year potentially causing a large shift of the maximum. As already discussed, this might also be connected to changes in precipitation distribution over the seasons."

**New reference:**

Sutton RT, Dong B, Gregory JM (2007) Land/sea warming ratio in response to climate change: IPCC AR4 model results and comparison with observations. Geophys Res Lett 34:L02701.

15. A couple of big concerns when reading the results section is about the structure, that I would like to hear from the authors. One is about the time periods, and the other is about the order of the presented results, both are connected on my question. Comparison of two

observational/historical periods (1951-1980) vs (1981-2010), and then future vs present periods (2071-2100) vs (1961-1990) are studied. In both cases, trends or differences are studied, and somehow compared. But both historical periods, purely in terms of temperature are not expected to present large changes, on the contrary to future scenarios analysis, and sometimes comparison of differences between both(e.g., 2oC, line 186 for historical periods, 10oC for future against present in line 201) makes a little bit confusing the interpretation of changes. Of course, differences on the first comparison is of 30 years, and it is of 90 years on the second comparison, when climate change signal is much more important. Perhaps directly study historical period comparing reanalysis against models, and then climate change signal would have been a more clear approach?.

Thank you to the reviewer for this question! Our intention with the two periods was to show both: (1) historical changes, which can be compared to observation-based reanalyses in order to illustrate where and to what degree the models are able to simulate historical seasonal changes. For this, we limited ourselves to the second half of the 20th century due to better data availability; (2) Projected changes, which we use to highlight the full extent of potential changes in a long-term view. With this, we aim to showcase instances where changes are getting quite profound, to illustrate potential impacts under a rather less optimistic socio-economic scenario.

16. My other concern is, related to this point, that if would have been a better results structure to first use the four indices for that historical period analysis (or the evaluation commented just before), and then the same for the climate change signal? How do the authors would feel about that view?

thank you for this suggestion. We originally also tried to more clearly separate the two time-periods in the description of the results, but this lead to quite some repetition and made a direct comparison of the results more difficult. In this sense, the results between the two historical periods and the changes projected for the end of the 21st century are inter-related to each other. Similar processes are supposedly behind the changes in both cases. Moreover, in many cases, the historical and future changes have the same character, just different magnitude. Therefore, we would prefer to leave the organization of the sections as it is. In several places in the section 4 we will add notes specifying whether that particular sentence is related to the historical or projected changes.

17. Several times, changes of the indices could be roughly related to global temperature trends or greenhouse gases rhythm of changes during the analysed periods. I guess if some idea about a potential relation with them could be added or commented during the analysis of the obtained results.

Thank you for this note. Does the reviewer refer to differences between the seasonally resolved regional results in comparison to annual global means here? This seems to be a good idea to better quantify what the effect of resolving the seasonal cycle can have on the estimation of changes.

We have added a table to the supplement giving the annual mean, global mean changes for both periods and all datasets (table S01) to enable a direct comparison between and refer to it in the discussion, when tackling the potential relationship of our results to climate sensitivity of individual ESMs:

"ESMs with higher climate sensitivity (corresponding to higher globally averaged temperature changes as listed in Table S01) generally project larger annual cycle shape changes (e.g., CanESM5 in the Arctic, Fig. 3, 4). Even though it has been argued that the higher sensitivities are not plausible (e.g., IPCC, 2021), it is difficult to rule out the hot models, especially in the case of regional impact assessment (Palmer et al., 2023; Swaminathan et al., 2024). This is illustrated in our results by cases where the regional changes in FDA diagnostics do not correspond to the differences in global mean temperature changes. For example, these cases include West Antarctica and regions in south-eastern north America."

18. Some comments are made about the precipitation role on the annual cycle over some parts of the globe (line 244), and so the dry/wet seasons more than the warm/cold seasons over extratropical regions. Do the authors think that perhaps over some regions the annual cycle of temperature is not very relevant, and so their potential changes?. The section referred to by the reviewer reads: "As the annual cycle in the near-equator regions is closely related to the seasonal distribution of precipitation, the shift of the maximum can indicate the change in the occurrence of dry and wet seasons. Over Africa, the first maximum is projected to occur earlier, and the second maximum is expected to be delayed. We note that in above mentioned regions with rather flat maximum and low amplitude, the ESMs and reanalyses mostly disagree on changes in amplitude."

Our aim here is to describe that in regions near the equator, the seasonal cycle of temperature is often less distinct than at higher latitudes, as no clear separation into winter/summer exists. At the same time, temperatures here might be (relatively) stronger affected by other climate system features, such as the Monsoon seasons or dry and wet periods in general. The smaller differences in temperature between seasons naturally lead to less of an effect of the annual cycle in absolute terms. However, the changes in the seasonal cycle, which we focus on here, might still be relevant (albeit smaller than, e.g., at midlatitudes). We have rephrased the section in question, to make this more clear, it now reads:

"As the (radiation-driven) annual cycle in the near-equator regions is less distinct, other climate system processes, such as the distribution of precipitation, become more important in shaping it. As a result, the shift of the temperature maximum can be indicative of a change in the occurrence of dry and wet seasons at low latitudes. At the same time, we note that in above mentioned regions with rather flat maximum and low amplitude, the ESMs and reanalyses mostly disagree on changes in amplitude, so that overall confidence in the signals is rather low."

19. One final remark: on line 358 it is said that many methods based on monthly variables, but most of the references studies seem to be using daily values, perhaps a more clear statement should be made when talking about other methods.

We are sorry for the confusion. We did not intend to say that most of previous methods used monthly data. We just wanted to emphasize the advantages of our method compared to using monthly or seasonal means.